# Modeling Phenological Phases across Olive Cultivars in the Mediterranean

**DOI:** 10.3390/plants12183181

**Published:** 2023-09-05

**Authors:** Ali Didevarasl, Jose M. Costa Saura, Donatella Spano, Pierfrancesco Deiana, Richard L. Snyder, Maurizio Mulas, Giovanni Nieddu, Samanta Zelasco, Mario Santona, Antonio Trabucco

**Affiliations:** 1Department of Agricultural Sciences, University of Sassari, 07100 Sassari, SS, Italy; 2Impacts on Agriculture, Forestry and Ecosystem Services (IAFES) Division, Euro-Mediterranean Center on Climate Changes (CMCC), 07100 Sassari, SS, Italy; 3National Biodiversity Future Center (NBFC), Palazzo Steri, 90133 Palermo, PA, Italy; 4Department of Land, Air and Water Resources, University of California, Davis, CA 95616, USA; 5Council for Agricultural Research and Economics, Research Centre for Olive, Citrus and Fruit Crops, 87036 Rende, CS, Italy

**Keywords:** phenological modeling, olive cultivars, phenological stages, the Mediterranean environment, CAC_GDD model

## Abstract

Modeling phenological phases in a Mediterranean environment often implies tangible challenges to reconstructing regional trends over heterogenous areas using limited and scattered observations. The present investigation aimed to project phenological phases (i.e., sprouting, blooming, and pit hardening) for early and mid–late olive cultivars in the Mediterranean, comparing two phenological modeling approaches. Phenoflex is a rather integrated but data-demanding model, while a combined model of chill and anti-chill days and growing degree days (CAC_GDD) offers a more parsimonious and general approach in terms of data requirements for parameterization. We gathered phenological observations from nine experimental sites in Italy and temperature timeseries from the European Centre for Medium-Range Weather Forecasts, Reanalysis v5. The best performances of the CAC_GDD (RMSE: 4 days) and PhenoFlex models (RMSE: 5–9.5 days) were identified for the blooming and sprouting phases of mid–late cultivars, respectively. The CAC_GDD model was better suited to our experimental conditions for projecting pit hardening and blooming dates (correlation: 0.80 and 0.70, normalized RMSE: 0.6 and 0.8, normalized standard deviation: 0.9 and 1.0). The optimization of the principal parameters confirmed that the mid–late cultivars were more adaptable to thermal variability. The spatial distribution illustrated the near synchrony of blooming dates between the early and mid–late cultivars compared to other phases.

## 1. Introduction

Countries around the Mediterranean basin have a principal and traditional role in olive production and its by-products. In fact, Spain, Italy, Greece, Turkey, Tunisia, and Portugal all together account for more than 95% of olive oil production worldwide [1].

Olive (*Olea europaea* L.) is one of the most long-lived tree crops, cultivated for thousands of years in the Mediterranean region. Its initial origin was identified in the eastern Mediterranean, before it expanded over the other parts of the Mediterranean basin in southern and southwestern Europe and northern Africa [2].

The olive tree’s growing area is mainly restricted to the region from 30° to 45° N [3]. This restriction suggests that climatic and particularly temperature conditions are the key factors driving and limiting olive growth processes. However, there are other ecological parameters affecting the suitability of different environments for olive growth, e.g., soil characteristics. An olive tree typically cannot withstand temperatures below −8 °C for more than one week, and high summer temperatures may also damage its yield performance [4].

Temperature acts as the main driver of olive tree phenology by regulating the release from the endo-dormancy period after the accumulation of adequate cold units during wintertime (chill units), and then the release from the eco-dormancy period, whose duration is dependent on forcing/heating units accumulated from the ending point of endo-dormancy to the bud breaking stage [5]. Hence, the breaking of winter rest and the onset of the pursuant vegetative stages are highly dependent on temperature [6]. Chilling and heating methods have presented different temperature ranges to account for chill and heat accumulation. Indeed, these considered both the lower and upper thresholds beyond which the temperature no longer affects the crop physiologically in terms of the growing and rest periods. However, divergences and uncertainty surround the required chill and heat accumulations and the period of time needed to complete phenological phases for different cultivars [7]. Flowering dates for olive were modeled in [8] considering olive development from the beginning of the season (i.e., the 1st of February) and the temperature sum reaching a defined number of growing degree days (GDDs) for the onset of flowering. On the other hand, GDD accumulation strongly depends on the specific olive cultivar and the initial date for heat accumulation.

In addition to temperature, there are genotype and several physical-environmental variables (e.g., distance from the sea, photoperiod, latitude, topography, and rainfall) that also influence olive blooming and other phenological stages [8,9,10,11,12]. Several authors have observed a delay in blooming with an increasing latitude and elevation and attributed this trend to the lower heat accumulation in the cooler zones [10,13,14,15]. According to Aguilera et al. [13], the earlier blooming dates of olive cultivars grown in southern Mediterranean regions (e.g., Tunisia) resulted from tree adaptation, a defense mechanism against temperatures above 30–35 °C occurring during the late spring months. Indeed, it was suggested that high temperatures are detrimental to the development and fertility of flowers [16].

Despite the varietal-specific phenological behavior (both in terms of timing and annual variability) of several tree species [17,18,19], only a few papers have proposed and compared phenological models for specific cultivars [8,20]. Several crop phenological models have already been developed and implemented by investigators in environmental and agricultural sciences to project phenological stages [18,21,22,23,24,25]. Previous available studies on olive phenology focused on modeling and estimating mainly the early phase in springtime (e.g., bloom). For instance, Rojo et al. [7] developed a study to define thermal accumulation to fulfill the chilling and heating requirements for the budbreak stage of olive trees in Toledo, central Spain; Lecce, southeastern Italy; and Chaal, central Tunisia. Considering both chilling and forcing requirements, they developed a phenological model that confirmed the highest performance in Toledo (with an error of about 2 days). Zouari et al. [20] applied the growing degree day (GDD) model to estimate the heating requirements of four olive cultivars for the flowering stage in southern Tunisia. They obtained a large inter-annual variation in GDD number while comparing differences between cultivars (from 100 to 267 GDDs). Orlandi et al. [26] developed a study to find linkages between olive flowering and heat accumulation using the GDD model in Mediterranean regions of Italy and Tunisia. The results confirmed that the olive species showed various heat requirements for flowering according to the latitudes of the experimental sites. Also, a phenological model was implemented for olive cultivars in Italy with a reverse-modeling approach using the developmental rate function, which was built on linear and nonlinear functions, to predict phenological stages from budbreak to complete flowering [27] for single locations.

The PhenoFlex model [24] was developed recently to predict the phenological stage of deciduous fruit trees in spring (e.g., budburst). It links both the dynamic model (for chill accumulation) and the growing degree hours model (for heat accumulation) with a very large number of parameters that offer high flexibility in simulating dormancy breaking. However, such an extensive parameterization usually requires a high number and many years of observations, which are not available for many sites.

The above-mentioned studies were mainly carried out separately at single locations that were characterized by specific environmental conditions and well-adapted local cultivars, and so the spatial projection of the phenological phases and validation over a wide region based on their results could have several limitations, since they did not cover a comprehensive range of environmental conditions.

Modeling the phenological stages of olive trees at regional or global scales (e.g., the Mediterranean basin) is highly challenging due to the limited and scattered observations across heterogenous environments. Some complex phenology models that are suitable for a single phenological stage, e.g., Phenoflex, which was built upon many parameters, and dependent on long-term phenological observations and hourly temperature time series would be unable to project comprehensively reliable phenological dates in the above-mentioned challenging experimental conditions. Thus, a general approach with a less parameter-demanding structure and flexible projection capability may perform better in modeling the successive phenological phases of the olive crop over a complex Mediterranean environment and consequently assist producers, particularly in a changing climate context, with crop cultivation management, planning crop practices in orchards, and mitigating climate-induced risks [28].

This study aimed to model and validate simulated dates for three main phenological phases, namely the sprouting, blooming, and pit hardening of early and mid–late budbreak olive cultivars, over the Mediterranean environment. To this end, we integrated observations from several experimental sites with different environmental characteristics in Italy and designed an innovative method to combine chilling and sequentially forcing for each phenological phase. Hence, to estimate the sprouting phenological phase, we first implemented the chill and anti-chill days model [18], and then, to estimate the blooming and pit hardening stages, we applied the growing degree days model [23] as a complementary method to accumulate only heat units beginning from the sprouting dates. Simultaneously, we also applied the PhenoFlex model, which is a complex process-based approach. The modeling performance of both approaches was compared by applying some statistics, including the root mean square error (RMSE), correlation, and standard deviation, according to the phenological phase and cultivar. The spatial implementation and projections of the modeling outputs were developed over Italy by phenological phase and cultivar type.

## 2. Materials and Methods

### 2.1. Data Collection

Phenological observations were obtained from different sources, including the PHENAGRI project (1996–2003) [29], the national network of CREA, and the Agricultural Department at the University of Sassari. We collected phenological data of four olive cultivars as the representatives of early and mid–late sprouting cultivars from nine experimental sites in Italy. The distinction of the two cultivar types was based on the observed mean phenological dates. The phenology date difference between our selected early and mid–late budbreak cultivars reached the peak at about 32 days for the sprouting phase, and then the pit hardening and blooming phases showed differences of about 28 and 10 days, respectively (Table 1). Figure 1 shows the geographical distribution of each experimental site in Italy. Most of these sites were good representatives of a wide range of Mediterranean climate types, spanning from the southern areas (Belice Mare, Sicily) with higher annual mean temperatures (18.9 °C), to the temperate climatic zone [30], i.e., the northern sites (Montepaldi and Sant Apollinare) characterized by lower mean annual temperatures of around 15 °C. The observation datasets in Julian day (JDay) format included three main phenological phases: sprouting, blooming, and pit hardening. The number of phenological observations/records to establish a timeseries list varied for each phenological phase and cultivar, e.g., from 11 to 32 records (Table 2).

For each year where the dates of phenological observations were available, the daily maximum, minimum, and mean temperatures were retrieved. The temperature timeseries were gathered from the European Centre for Medium-Range Weather Forecasts (ECMWF) Reanalysis v5 (ERA5) [31], since weather stations were not available near all the experimental sites. Also, some of them provided fragmentary timeseries. Indeed, the use of reanalysis products facilitates the parameterization and validation of model assessments more closely linked to climate model products employed for regional projections. Meanwhile, a previous study [32] found that olive phenological modeling using the ERA5 reanalysis temperature data series provided fairly accurate phenological projections. This dataset was based on the hourly ECMWF ERA5 reanalysis data at 2 m above surface level with a horizontal resolution of 0.1° and aggregated at a daily temporal scale.

Figure 2 shows the dispersion of daily mean temperature between all experimental sites and the selected years for which phenology was monitored. The temperature distribution confirmed the lowest and highest median ranks in February (6.7 °C) and August (27.5 °C), respectively. During winter and summer, the temperature percentiles (25th–75th and 5th–95th) were more stretched, which indicated higher temperature variability between the experimental sites in the cold and warm seasons. Figure 2 also shows box plots of the distributions of phenological dates from the experimental sites for sprouting (A), blooming (B), and pit hardening (C).

### 2.2. Phenological Models

The chill and anti-chill days model (CAC model), or chilling and forcing model [18], is a sequential model that accumulates chill days until the chill requirement is fulfilled and endo-dormancy is over. Then, the accumulation of anti-chill days starts during the eco-dormancy stage to overcome the quiescence. Indeed, the dormancy stage is divided into two phases: (1) the endo-dormancy phase, in which the plant reaches the peak of chilling accumulation once meeting its chill requirements, and (2) the eco-dormancy phase, in which the crown buds are in suspension and their growth is influenced by environmental factors, i.e., temperature/heating [33].

To calculate chill days (Cd) and anti-chill days (Ca), we implemented a set of equations using the single triangle degree day computation method for different temperature conditions (Table 3). In the original reference (citation), there were five cases, to which we added one more based on temperature data peculiarity (i.e., max daily temperature below 0 °C). Indeed, to calculate chill and anti-chill days, this model included two basic parameters: a temperature threshold (Tc) and chilling requirement (Cr). To find the best values or optimize these parameters, we developed an error function in the R computer language (version: 2022.02.3+492) to employ the fitness function of the genetic algorithm (GA) package [34]. A GA was implemented for stochastic optimization and to optimize the provided error function in terms of fitness, binary, real-valued, and premutation representations, which were available in the package. This package needed some inputs: temperature timeseries (min, max, mean); phenological observation data; and the pre-selected lower and upper bounds of the parameters (Tc: 7–14, Cr: −80–−200). We changed these bounds in the optimization process to find the lowest possible errors. After arranging all inputs in the code, we set the number of iterations to 1000.

Since the CAC model was developed to estimate only the first phenological stage, the end of dormancy, a combined method of the chill and anti-chill days model + the growing degree days model (CAC_GDD model) was developed to include and estimate blooming and pit hardening stages. If the CAC model was used to assess two or three consecutive phases of a particular cultivar, different chill requirements per phase would be accounted for, which is per se against the principles of crop physiology. Indeed, the starting dates to accumulate heating would be different spatially (point by point) and temporally (year by year), and thus we present a dynamic method for GDD accumulation. The main parameters that the GDD model requires to calculate heat accumulation for subsequent phenological stages include the temperature base (Tb), maximum temperature base (Tx.base), and heating requirement (Hr).

The GDD equation is as follows:GDD = (Tx + Tn)/2 − Tb 
where Tx is the daily maximum temperature, Tn is the daily minimum temperature, and Tb is the base temperature. The daily minimum and maximum temperatures should be set to Tb if less than Tb and set to an upper temperature threshold (Tx.base) when greater than that threshold, because most plants cannot grow efficiently beyond these thresholds [23]. To run the calibration using this method, an error function was again developed based on the GDD model to include in the GA function in order to optimize the parameters. The inputs included temperature timeseries (min, max); phenological observation data; sprouting observation data, such as the pervious phenological date; and the pre-selected lower and upper bounds of the three above-mentioned parameters (i.e., Tb: 4–10, max Tb: 25–35, Hr: 300–1400).

The PhenoFlex model was based on the structure of the dynamic model and the growing degree hours model (GDH) for chilling and heating accumulations [24]. This model fit the main parameters of both the dynamic and GDH models for the phenological observation dates. A generalized simulated annealing algorithm (GSA) was applied to calibrate the model. The principal parameters to fit the PhenoFlex model were as follows:yc—Chilling requirement: critical value of y, which defines the end of chill accumulation.zc—Heating requirement: critical value of z, which defines the end of heat accumulation.s1—Slope parameter that determines the transition from the chill accumulation to the heat accumulation period in PhenoFlex.Tu—Optimal temperature of the growing degree hours (GDH) model.E0—Time-independent activation energy of forming the precursor to the dormancy-breaking factor (PDBF).E1—Time-independent activation energy of destroying the precursor to the dormancy-breaking factor (PDBF).A0—Amplitude of the (hypothetical) process involved in forming the precursor to the dormancy-breaking factor in the dynamic model.A1—Amplitude of the (hypothetical) process involved in destroying the precursor to the dormancy-breaking factor (PDBF) in the dynamic model.Tf—Transition temperature parameter of the sigmoidal function in the dynamic model, also involved in converting the PDBF to chill portions.Tc—Upper threshold in the GDH model.Tb—Base temperature of the GDH model.slope—Slope parameter of the sigmoidal function in the dynamic model, which determines what fraction of the PDBF is converted to chill portions.

Thus, the PhenoFlex package [35] needs several inputs to fit the model parameters with observations, including: the observed dates of the desired phenological phase; a function called PhenoFlex_GDHwrapper, which uses a heating model considering the GDH concept; a season-based timeseries of maximum and minimum temperatures in an hourly scale corresponding to the observed phenology years; and, finally, the default initial estimates and upper and lower changeable bounds of the 12 aforementioned parameters.

### 2.3. Data Preparation

Daily temperature data series from the ERA5 repository were extracted as NetCDF over a spatial window (5–20° E and 35–47° N) at a spatial resolution of 0.1 degree for the period 1985–2015, with an additional list of yearly vectors (November of the previous year to October) for the coordinates of each experimental site. For the basic processing of the temperature data series, the Climate Data Operator (CDO) was used. Observed phenological data series per cultivar and phenological stage were imported in R as vector lists. We made data frames for all weather and phenological observation data series to use in the model fitting functions.

### 2.4. Calibration and Validation of Phenological Models

We developed error functions for the CAC model and GDD model separately. CAC’s error function worked based on the computation method of chill and anti-chill units. Still, the growing degree days error function was built on the GDD function in the pollen library [36] using the sprouting dates as the starting point. Both developed error functions were nested in the GA function to optimize the parameters. Providing all above-mentioned parameters and inputs with each of the models using the maximum number of iterations of the algorithm (i.e., 1000), the GA function found the best-fitted parameters with the lowest RMSE.

The PhenoFlex model fit the data based on the generalized simulated annealing algorithm (GSA). To run the phenology fitter, we used seasonally arranged weather timeseries (November–October) and phenological observation data series. Daily maximum and minimum temperature timeseries were converted to hourly timeseries based on the idealized daily temperature curve presented by Linvill [37]. The number of iterations was set to 1000, with five search steps in the algorithm (as recommended in the default).

To validate the results, we performed a leave-one-out cross-validation (LOOCV), which is considered suitable when the number of observations is limited [38]. We coded and applied an LOOCV in R to calculate the RMSE values from both approaches (CAC_GDD and PhenoFlex) per cultivar and phenological phase. LOOCV iteratively uses one observation to test the performance of the model (i.e., the RMSE) calibrated with all the remaining observations. Thus, the mean and standard deviation of these values were used to assess the strength and variability of the performance of each model.

After obtaining the calibration and cross-validation results for each phenological model, the observations versus estimated values were statistically tested using the root mean square error (RMSE) and coefficient of correlation. Furthermore, a Taylor diagram was plotted to show additional measures of model performance by phenological stage comparing the modeled and observed values. The phenological dates were spatially projected using the CAC_GDD model, focusing on Italy. For the spatial implementation, a mapping code was developed in R, and then the optimized parameters were used along with the long-term mean daily temperature timeseries over 30 years (1985–2015) from the ERA5 repository with a spatial resolution of 0.1 degree over a window of 5–20° E and 35–47° N, including Italy.

## 3. Results

The RMSE values obtained from the calibration of the CAC_GDD model by phenological phase showed the lowest errors for the blooming (4–12 days) and then pit hardening (5–13 days) stages, while the errors were approximately doubled (6–24 days) for the estimates of the sprouting stage. The PhenoFlex model found similarity with the chill and anti-chill days model in the sprouting phase, with errors ranging from 5 to 24 days. However, PhenoFlex produced larger errors in the blooming (14–18 days) and pit hardening (13–42 days) stages. Comparing the models’ functionality based on the cultivars, higher errors were generally observed for early-budbreak representatives (i.e., Carolea and Picholine) regardless of the phenological model for all three phenological phases, ranging from 9 days for the CAC_GDD model in the blooming phase to 42 days for the PhenoFlex model in the pit hardening phase. In contrast, the estimated errors were lower for the mid–late budbreak cultivars, Frantoio and Moraiolo, ranging from 4 days for the CAC_GDD model in the blooming phase to 23 days for the PhenoFlex model in the pit hardening phase (Table 4).

The principal model parameters to estimate phenological dates were optimized by phenological phase, cultivar, and phenological model (Table 5). The use of the CAC model to estimate sprouting dates suggested that the best temperature thresholds (Tc) for the early budbreak cultivars ranged from 8.2 to 9.5 °C, whereas for the mid–late budbreak cultivars they ranged from 9.7 to 11.2 °C. The optimized chill requirements (Cr) for early budbreak cultivars ranged from −115 to −122 chill units, while for the mid–late representatives the range was −133 to −137 chill units. The CAC_GDD model optimized three parameters, including base temperature (Tb), maximum base temperature (Tx), and heat requirements (Hr), to estimate both the blooming and pit hardening dates. The derived best Tb and Tx parameters ranged from 4.5 to 7 °C and 26.3 to 31.5 °C, respectively, and varied according to phase and cultivar. The optimized Hr parameter presented higher values for early compared to mid–late budbreak cultivars in the blooming phase (437–568 GDD° versus 370–388 GDD). Using the same calibrated Tb and Tx of the blooming stage, we tried to optimize the Hr for the pit hardening phase, but then the mid–late cultivars obtained higher heat requirements (1275–1315 GDD versus 1003–1073 GDD).

The results of the leave-one-out cross-validation (Table 6) verified the model calibration. As expected, this validation showed similar results for the CAC and PhenoFlex models in the sprouting phase, particularly for the late cultivars, by indicating mean RMSEs ranging from 5.74 to 11 and 5.65 to 12.48 days, respectively. However, the differences between the two phenological approaches increased for blooming (CAD_GDD: 3.47–12.23 days, PhenoFlex: 29.9–36.4 days) and grew substantially for the pit hardening phase (CAD_GDD: 4.5–14 days, PhenoFlex: >40 days). In regards to the standard deviation values of the RMSEs obtained through cross-validation, the PhenoFlex model indicated the highest variability, from 0.5 to 5.87, versus CAC_GDD, with a range of 0.4–1.4.

Using the optimized parameters and temperature timeseries, the modeled dates and phenological phases were determined for different cultivars. Figure A1 (Appendix A) shows the accumulation of chill and anti-chill units for the sprouting stage and the GDD accumulation for both the blooming and pit hardening stages as an example for the Carolea cultivar over the years with available phenological data for different sites. After the fulfillment of the chill requirements (i.e., −115 chill units), anti-chill units accumulated until zero (i.e., the chill units balanced out), at which point in time sprouting occurred. For the blooming and pit hardening dates, GDD units accumulated until the heat requirements were fulfilled (e.g., 437 GDD for blooming and 1074 GDD for pit hardening).

The estimated dates for all phenological phases and cultivars were compared to observations. The CAC_GDD model values showed significant correlation coefficients (*p*-value < 0.05) for the three phenological phases (0.55, 0.71, and 0.80, respectively), higher than for the bloom and pit hardening phases estimated from the PhenoFlex model (0.53, 0.28, and 0.49, respectively by phenological phase) (Figure A2 (Appendix B)).

The estimated phenological dates considering both models exhibited the lowest variability in the blooming phase during the spring season, coping with a narrower distribution of temperature and observed phenological dates, as shown in Figure 2. For both models, the estimated sprouting dates had distributions more similar to the observations, or a higher predictability, than for blooming or pit hardening (Figure 2 and Figure 3): the median values of the sprouting stage (observation = 12 April, CAC = 15 April, and Flex = 20 April) versus the median values of the blooming stage (observation = 25 May, CAC_GDD = 29 May, Flex = 15 May) and the median values of the pit hardening stage (observation = 15 July, CAC_GDD=17 July, Flex = 10 June). Using a normalized Taylor diagram, the applied phenological models’ performances were compared through the standard deviation, RMSE, and correlation coefficient (Figure 4). For the experimental results and data available in our study, the CAC_GDD models showed higher consistency with observed phenological dates than the PhenoFlex model for the blooming and, in particular, pit hardening phases. In particular, CAC_GDD showed higher correlation coefficients compared to the PhenoFlex model (i.e., blooming: 0.7 vs. 0.28, pit hardening: 0.8 vs. 0.49); a lower RMSE (i.e., blooming: 0.8 vs. 1, pit hardening: 0.6 vs. 1.3); and standard deviations closer to the observations (i.e., blooming: 1 vs. 0.5, pit hardening: 0.9 vs. 1.5). For the sprouting phase, both models indicated similar behavior and predictability skills (r = 0.54, Sd = 0.4, and RMSE = 0.9).

To verify the model’s functionality for spatial distribution, we implemented the CAC_GDD model over Italy using ERA5 reanalysis data for the 1985–2015 period, which was evaluated as providing better performance under the available training conditions. The spatial patterns (Figure 5) clearly identified later estimated phenological development dates under a colder climate, i.e., higher latitudes and mountainous regions, and earlier estimated dates over southern regions and lowlands for any phenological phase or cultivar type. For the sprouting, blooming, and pit hardening phases, the estimated dates over highlands (i.e., mountainous areas) and northern regions showed JDay values >130, >160, and >210, respectively. However, the aforementioned phenological dates were the latest estimates in the southern regions and lowlands, e.g., around coastal areas. Comparing the estimated dates of the late and early budbreak cultivars (diff = late − early) over the study area, we mostly found differences greater than 15 days for the sprouting and pit hardening stages. Still, a smaller difference was observed for the blooming stage, as the late cultivars reported general delays of less than 15 days compared to the early cultivars.

## 4. Discussion

The present study compared two approaches to derive and evaluate estimates of olive phenological phases by applying chilling- and forcing-based models, which could be inferred from scattered monitoring over a large Mediterranean environment. Our results suggested that generalized projections of olive phenology might be possible under the available modeling setup, especially with the CAC_GDD model, which could support the strategic management of olive cultivation and the anticipation of long-term changes (e.g., under climate change projections) for more structural adaptation at the regional scale.

Overall, the CAC_GDD approach, which was based on both the single triangle method [18] and heat accumulation [23], provided more feasible results under our scattered experimental setup than the PhenoFlex model [24], which was based on the dynamic model [39] and the growing degree hours model [40,41]. The CAC_GDD approach required fewer parameters for calibration while avoiding overfitting, especially with inadequate data. The present approach used daily climate data, preventing the implementation of artifacts to transform them into hourly climate data. In addition, CAC_GDD reported fewer model projection failures under the wide climatic range in a large-scale spatial implementation, and the approach required less computational effort and processing time, which might facilitate its implementation on a large scale for management tools. PhenoFlex provided a more complex process-based representation considering many parameters but also required more processing time and suffered to a certain extent from model assessment failure in a comprehensive spatial implementation. PhenoFlex is an open-source model that can flexibly adapt to various species and cultivars based on its strong biological and experimental structure for dormancy dynamics. This integrated model may easily outperform other models in reconstructing complex phenology-related dynamics at the local scale and especially in relation to dormancy [24]. However, such a detailed workflow with a large number of parameters and degrees of freedom may undermine the more articulated and accurate representation of phenological phases by PhenoFlex given the more limited and scattered number of observations available, which is unfortunately often the status of many tree crops in Mediterranean areas, and thereafter compromise the feasible projections at the regional scale.

The RMSE values indicated the better performance of the CAC_GDD model for the blooming stage of the mid–late budbreak cultivars, while PhenoFlex performed best for the sprouting stage of the mid–late cultivars. Both models performed better for the mid–late budbreak cultivars than the early ones, regardless of the phenological stages. These findings suggest that the cultivars differed in their sensitivity to weather conditions, with the mid–late cultivars being more sensitive. In the framework of climate change and the increasing uncertainty of weather events, the possibility of selecting between cultivars characterized by different degrees of sensitivity to seasonal weather might be a useful decision-making tool for producers. The best performance of the phenological model was found for the blooming stage of the mid–late budbreak representative cultivars through our approach, with an error of around 4 days. The reason behind this finding was likely that the mid–late cultivars showed similar behavior or parameters in terms of the threshold temperatures (Tc, Tb, and Tx) and Hr. In contrast, Carolea and Picholine, selected as early budbreak cultivars, showed differences between each other. These findings highlight the importance of considering the varietal factor to create reliable and reproducible phenological models. Moreover, it is worth noting that Moraiolo and Frantoio, as mid–late cultivars, are native to the same area (Central Italy), while this was different in the case of Carolea (South Italy) and Picholine (France) [42,43]. These findings encouraged us to formulate the following hypotheses: (1) cultivars from similar historical growing areas with similar environmental conditions have developed common adaptive capacities and features; and (2) environments characterized by several limiting factors for olive tree species’ growth, related to both extreme temperatures and short-term periods suitable for achieving certain critical phenological phases (e.g., blooming), might have selected cultivars that present high sensitivity to temperature changes and thus allow better predictions. However, these hypotheses were based on limited data from just four cultivars; the enlargement of the study to other cultivars is needed to verify these hypotheses and investigate possible common varietal patterns.

The CAC_GDD model calibrated for each distinct cultivar resulted in optimal temperature thresholds that aligned with their early and mid–late phenological behavior. The optimized base temperature (Tc) and the corresponding chilling requirements (Cr) were lower for the early cultivars as opposed to their counterparts. The Tc and Cr values derived from the models underscored the substantial distinctions among the cultivars, with differences of about 3 °C (e.g., Tc = 8.2 °C for Picholine compared to Tc = 11.2 °C for Moraiolo) and approximately 20 chill days (e.g., Cr = −115 for Carolea compared to Cr = −137 for Frantoio). Given the projections of escalating temperatures in future climate scenarios, particularly impacting the winter seasons, coupled with the fact that distinct cultivars exhibited specific threshold temperatures and chill prerequisites, challenges pertaining to the productivity of traditional cultivars and their corresponding geographical domains are likely. Notably, a significant number of these well-established cultivars have evolved over centuries through selective breeding and adaptation to specific microclimates [43]. Furthermore, the determination of such critical values will assume even greater significance in aiding producers’ decisions regarding the most appropriate cultivars in response to anticipated shifts in climatic conditions. Analogous observations could potentially extend to threshold temperature parameters (Tb and Tx) for heat accumulation, which in this context could increase concerns of risks such as late frost events or heatwaves coinciding with critical flowering phases.

Our obtained RMSE values were in accordance with those of some previous publications, e.g., [8,13,27,44], which showed errors of modeling for the blooming/flowering stage of olives ranging from 3 to 8 days in different areas.

The results obtained by Cesaraccio et al. [18] in estimating the budbreak/sprouting stage using the chill and anti-chill days model for the olive crop in Oristano (Sardinia) showed an error of about 8 days, which could approximately confirm (of course, depending on the cultivar) our obtained RMSEs for the sprouting stage, which ranged from 6 to 24 days, as we found lower errors (6–9 days) for mid–late and higher errors (20–24 days) for early budbreak cultivars.

Overall, the cross-validation confirmed the validity of the calibration results with much smaller differences for the CAC_GDD model than PhenoFlex, which showed extremely high RMSEs, particularly over the pit hardening phase. These large differences might have been due to the method, as PhenoFlex accumulated heating and chilling starting from November first to predict pit hardening dates in the summer. This represented a longer prediction period, using greater time spans where the model could be particularly prone to errors. Notably, PhenoFlex is a model that is basically qualified to project only a single phenological phase, e.g., flowering [24], and it has not yet been adapted to work over successive phases. In contrast, CAC_GDD accumulated heating to predict the pit hardening stage beginning from the already estimated sprouting dates and using the new parametrizations based on the growing degree days model, and accordingly it avoided high errors with consecutive phenological phases.

Comparing the functionality of the above-mentioned approaches in terms of the prediction of the three phenological phases, we found that CAC_GDD demonstrated better modeling performance for two phases, namely pit hardening and blooming, while for sprouting the PhenoFlex and CAC_GDD models were quite similar, with relatively poor modeling performance. We could then refer mainly to the blooming phase to more accurately assess the performance results. Most previous phenology studies have projected and focused on olive’s flowering/blooming stage, as reported in the following examples. The findings of de Melo-Abreu et al. [8], using a thermal time method to estimate flowering dates, showed an RMSE range of about 2–5 days and a margin of error of 0.57–0.74, indicating acceptable model performance. Rojo et al. [7] predicted olive pollination dates (which would be considered to represent the flowering stage) in Toledo (Spain), Lecce (Italy), and Chaal (Tunisia) using chilling and heating accumulations; the results displayed a mean absolute error of about 4.5 days on average, which confirmed the accurate prediction of the flowering stage. Moriondo et al. [22], using the Unichill model, demonstrated a good simulation of flowering with RMSEs of around 3 and 3.8 days for calibration and validation, respectively. Although the previous studies indicated lower errors, they were developed and established over separate single sites. Hence, these model assessments were not suitable to project phenological phases over a large area for a spatial analysis, while our approach tried to consider an extensive range of environmental and bio-physiological conditions combining observations from eight experimental sites in Italy.

The optimization of the principal parameters to estimate sprouting dates suggested higher Tc (average ~10 °C) and Cr (average ~−135 chill units) values for the mid–late budbreak cultivars than for the early representatives (Tc: average ~8.8 °C, and Cr: average ~−118 chill units). On average, we found differences of 1.2 °C in the Tc and −17 chill units in the Cr. Indeed, the temperature thresholds and chill requirements optimized by the chill and anti-chill days model varied by cultivar type. Similar results, particularly considering our findings for mid–late olive cultivars, were reported by Orlandi et al. [45] for threshold temperature estimation (6–12 °C) and by Cesaraccio et al. [18] for threshold temperature (10.6 °C) and chill requirements (−138 chill units). The higher Tc and Cr values of the mid–late cultivars suggest an adaptation feature to avoid early spring frosts, consistent with the typical environmental conditions of the Frantoio and Moraiolo areas of origin [30].

Using the CAC_GDD model, we optimized the principal parameters for the blooming and pit hardening stages. The observed Tb (average 6.5 °C) and Tx (average 29.9 °C) values for early budbreak cultivars were higher than those estimated for the mid–late representatives (Tb: average 4.9 °C, and Tx: average 27.9 °C) with a difference of around 1.6 °C in Tb and 2 °C in Tx. The heat requirements optimized to represent the blooming stage for early cultivars were about 128 GDD higher than for the mid–late ones (means of 502 versus 379 GDD). As the GDD accumulation started from sprouting, the mid–late budbreak cultivars with delayed sprouting dates accumulated heating later than the early ones and needed fewer GDD till blooming. On the other hand, for the pit hardening stage, the early budbreak representatives showed lower heat requirements than the mid–late sprouting cultivars (i.e., means of 1038 versus 1295 GDD). Therefore, the pit hardening event also occurred earlier for them. Previous studies about heat requirements, e.g., [8,10] confirmed our results, indicating a range of 180–560 GDD in the heat requirements for the flowering of olive cultivars and Tb values from 5 to 12.5 °C. Using a machine learning model, Oses et al. [46] found base temperatures below 10 °C, similar to our optimized Tb values, to predict olive phenology, resulting in better model performance. Notably, the above-mentioned threshold values varied with the experimental sites’ climatic characteristics and the cultivar type. Warmer regions and early budbreak cultivars could have a higher base temperature. In contrast, cooler regions and late budbreak cultivars, e.g., highlands and higher latitudes or late budbreak cultivars, could have a lower base temperature. Consequently, differences in the estimated principal parameters could depend on bio-geographical characteristics [10]. Considering the optimized Tb values, the mid–late olive cultivars seemed more resistant under cold climates. Bio-physiologically, temperature changes could disturb olive’s phenological process, quality, and yield, as low temperatures could cause bark cracking and the death of thick branches. On the other hand, high temperatures shrink the fruit body and its pulp [47].

Because of the lower temperature variability during the blooming season between the experimental sites in Italy, the observed and modeled dates were similar, with a mean range of about 21 days between the 25th and 75th percentiles. Therefore, with the increasing temperature variability during the sprouting and pit hardening phases (cold and warm seasons, respectively), the corresponding range of phenological dates was stretched, with a mean range of about 35 days for both sprouting and pit hardening between the 25th and 75th percentiles. di Paola et al. [27] found a similar consistency between temperature and olive phenological date distributions in Italy. Despite the higher temperature variability in the warm season, the CAC_GDD model demonstrated the best performance for the pit hardening phase regardless of the cultivar type. Due to the limited number of previous studies on phenological phases other than blooming, these findings may be particularly relevant to improve estimates of the pit hardening phase, which would be applicable for different olive cultivars and other species. Pit hardening is considered a critical phase of olive fruit development that usually corresponds to the end of the first phase of fruit growth, which is characterized by intense cell division and the sinking of assimilates in endocarp tissues [48]. This indicates the beginning of oil accumulation in the fruit [49]. Forecasting the pit hardening date is a useful tool for irrigation management [50], the application of phytosanitary products [51], the further estimation of the peak of oil accumulation, and the determination of the optimal harvest period [49].

The CAC_GDD’s implementation and projections were also verified spatially over Italy. All three phenological phases showed the earliest estimated dates under warm climates, including southern regions and lowlands (e.g., around coastal areas). However, the late phenological dates were estimated over mountainous areas (e.g., the Alps and Apennine mountains). This result demonstrated the relationship between spatial changes in temperature and the fulfillment of different cultivars’ heat requirements and, consequently, olive phenological dates [7]. In summary, in a warmer climate, forcing units or growing degree days accumulate faster to reach the required threshold than in cold areas. However, the chilling requirements can also be met earlier in a warm climate if daily temperatures do not exceed Tc during endo-dormancy. In some hyper-cold regions (i.e., in the Alps), the model failed to accurately estimate the phenological dates. This result was likely due to the lower temperatures, i.e., below the temperature thresholds, which prevented the model from accumulating adequate chilling or heating to fulfil the defined requirements. Consequently, no phenological dates were obtained.

The spatial results confirmed that the model produced smaller differences between the blooming dates of the early and late cultivars than for both sprouting and pit hardening phases over most of the study area, i.e., less than 15 versus more than 15 days. This projection was likely due to the optimized parameters of the CAC_GDD varietal model. Indeed, in warmer climates, sprouting advanced, and the differences in phenological dates between the two varietal types were smaller. Moreover, the earlier sprouting dates occurring during the year with a lower probability of days with temperatures exceeding the Tx.base thresholds and lower Hr requirements could advance the blooming dates for the mid–late cultivars. Relative to projected climate change, a marked advance in the blooming date caused by warming would be beneficial, since it might avoid production losses due to flower damage caused by heat waves [52], particularly in a warm climate. However, the preliminary nature of these spatial projections would encourage a new investigation with more data from more cultivars in olive-growing regions (e.g., North Africa) to improve parametrization and validation.

## 5. Conclusions

Two approaches applying chilling- and forcing-based models were compared to determine the performance for predicting olive phenology. This investigation presented a combined method (CAC_GDD) to estimate olive phenological phases and compared results using a more complex and data-demanding model (PhenoFlex). Under our experimental conditions, over a large Mediterranean environment with scarce and scattered observations, the CAC_GDD model, with a lower parameter demand and simple approach, demonstrated more reliable performance than the PhenoFlex model to generalize projections at the regional scale in at least two phenological phases, i.e., the blooming and pit hardening stages. However, in terms of cultivar type, both models performed better for the mid–late than the early budbreak cultivars.

CAC_GDD showed some advantages for modeling the phenological phases. For example, CAC_CDD (1) required less parameterization; (2) used daily temperature timeseries with no artifacts, i.e., no transformation into hourly data; (3) needed less computation effort; (4) demonstrated faster processing; and (5) reduced model projection failures in a spatial implementation.

Considering the bio-geographical characteristics that determined the temperature thresholds, i.e., the heat and chill needs for each species, the principal parameters optimized through the present approach showed clear differences by olive cultivar type. The mid–late budbreak representatives were more adaptable than early cultivars to the cold climate when considering sprouting. Still, they could also adapt to warmer climates by anticipating earlier blooming dates.

For the model calibration and validation, the present investigation considered a more comprehensive range of environmental and bio-physiological conditions combining phenological observations from nine experimental sites. The CAC_GDD model could project phenological phases over a large area, demonstrating the spatial pattern of olive phenology. The model only failed to accurately project phenological dates over very high areas (i.e., in the Alps mountain region). Indeed, the model failures occurred only when the daily maximum and minimum temperatures were beyond the thresholds. These conditions prevented the model from accumulating sufficient chilling and heating units to meet the requirements. From a spatial point of view, we found smaller differences in phenological dates between the early and mid–late cultivars for the blooming phase over most parts of the study area, revealing the higher phenological plasticity of the mid–late cultivars.

Our approach will support olive producers’ responses to future climate change through the resilient strategic management of olive cultivation, varietal choice, cultural practices, and mitigating climate-induced risks based on the reliable projections of different phenological dates. As a future research direction, the CAC_GDD model could support spatial upscaling to a large region, e.g., the Euro-Mediterranean region, in order to display the environmental differences in phenological projections under future climate change scenarios. Nevertheless, collecting more phenological observations for additional olive cultivars and other olive-growing regions would support our approach to producing more integrated and comprehensive validation results and promote our model’s performance.

## Figures and Tables

**Figure 1 plants-12-03181-f001:**
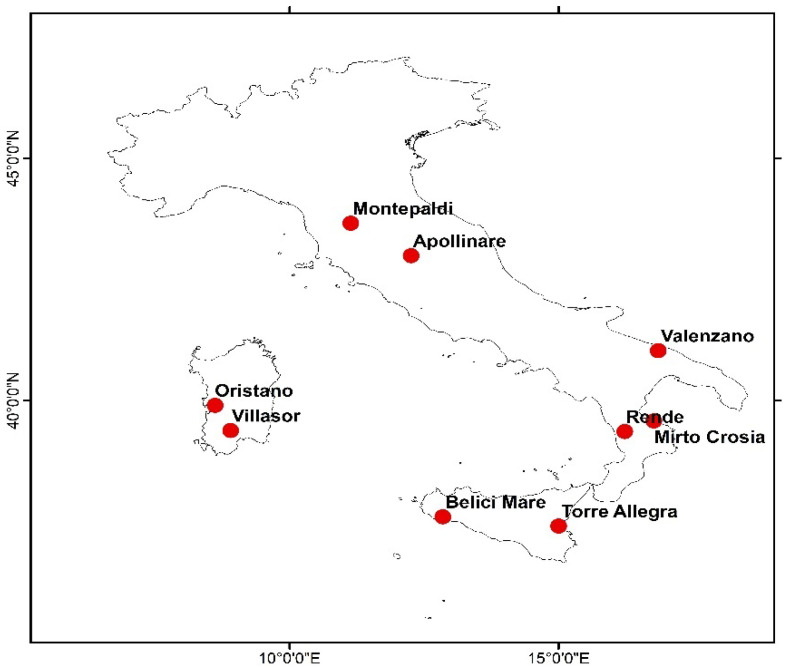
Experimental sites in Italy with olive phenological observations and associated dates.

**Figure 2 plants-12-03181-f002:**
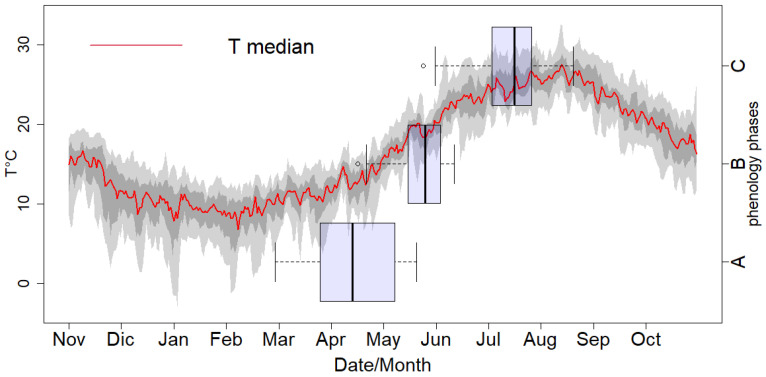
Mean and range of daily temperatures for those years with available observations from the experimental sites. Red line is the median, dark grey shading indicates the 25th to 75th percentile, and light grey shading indicates the 5th to 95th percentile. Boxplots show the dispersion of the observed phenological dates (JDay) per phase, including sprouting (**A**), blooming (**B**), and pit hardening (**C**) between all experimental sites, cultivars, and available years. The median line is in the center, and the boxes and the whiskers extend from the 25th to the 75th percentile and from the 10th to the 90th percentile, respectively.

**Figure 3 plants-12-03181-f003:**
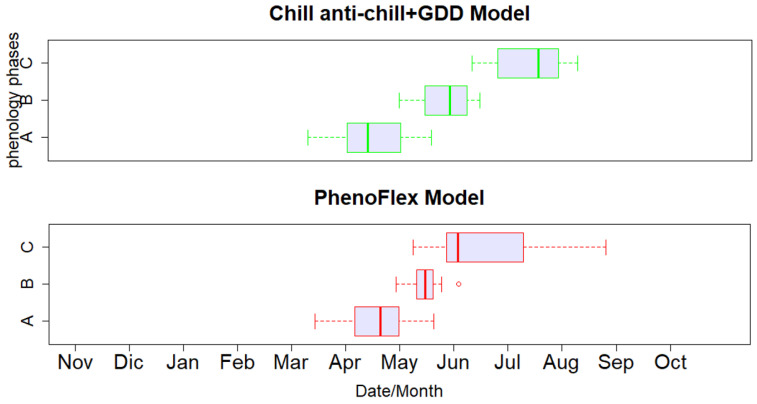
Boxplots showing distributions of phenological dates estimated by the CAC_GDD model (top plot) and the PhenoFlex model (bottom plot) for all olive cultivars and phenological stages, including (**A**) sprouting, (**B**) blooming, and (**C**) pit hardening.

**Figure 4 plants-12-03181-f004:**
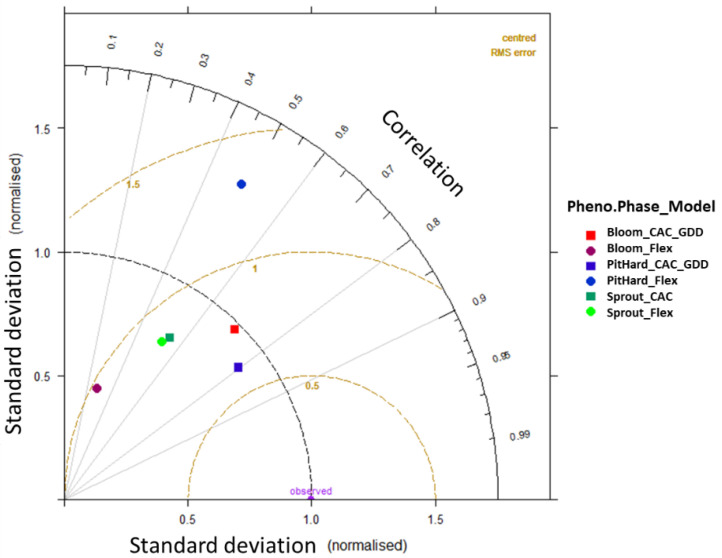
Normalized Taylor diagram showing CAC_GDD and PhenoFlex phenological models’ performances in estimating the selected olive cultivars’ phenological dates.

**Figure 5 plants-12-03181-f005:**
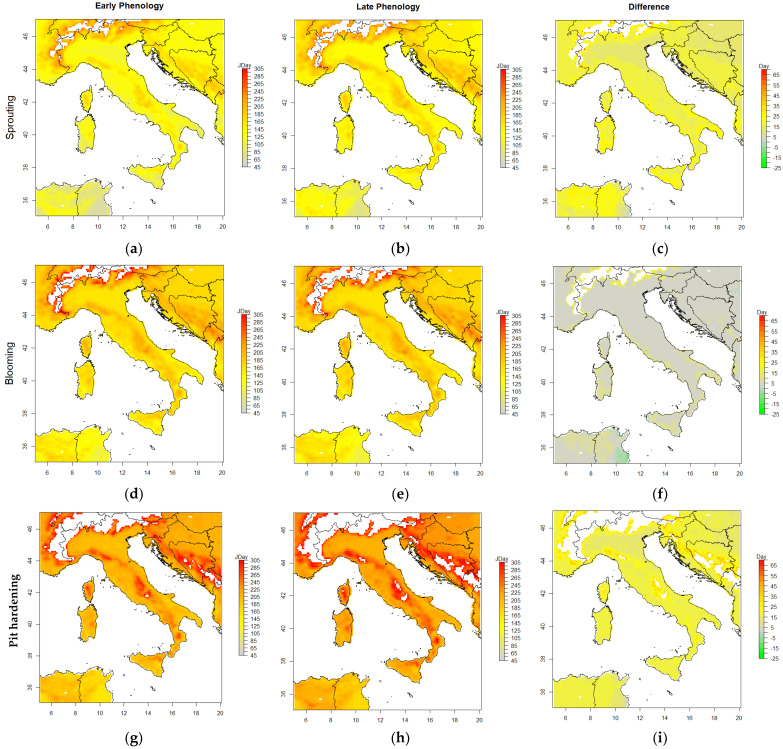
Spatial implementation of the CAC_GDD model to estimate olive phenological phases/dates (in JDays), including sprouting (**a**–**c**), blooming (**d**–**f**), and pit hardening (**g**–**i**) for early (**a**,**d**,**g**) and late (**b**,**e**,**h**) budbreak cultivars over Italy. The right column (**c**,**f**,**i**) shows the spatial difference between late and early cultivars (diff = late − early) for estimating phenological dates. Calculations were carried out with long-term daily mean temperature timeseries from ERA5 over a 30-year historic period (1985–2015).

**Table 1 plants-12-03181-t001:** Mean observed dates (in JDay) of investigated phenological phases for selected early and mid–late budbreak olive cultivars.

Phenological Phase Cultivar	Sprouting	Blooming	Pit Hardening
**Carolea/early**	96 (6 April)	138 (18 May)	186 (5 July)
**Picholine/early**	97 (7 April)	140 (20 May)	187 (6 July)
**Frantoio/mid–late**	124 (4 May)	148 (28 May)	214 (2 August)
**Moraiolo/mid–late**	134 (14 May)	151 (31 May)	215 (3 August)

**Table 2 plants-12-03181-t002:** Experimental sites in Italy, with available phenological monitoring and number of years of phenological observations for early (i.e., Carolea and Picholine) and mid–late (Frantoio and Moraoiolo) olive cultivars.

Location/Site	Latitude (Decimal Degrees)	Longitude (Decimal Degrees)	Tm	Cultivar	Years of Data Availability	Length (Years)
**Montepaldi (Tuscany, FI)**	43.66°	11.14°	15.8 °C	Carolea	1997–1999	3
				Picholine	1997–1999	3
				Frantoio	1997–1999	3
				Moraiolo	1997–1999	3
**Villasor (Sardinia, CA)**	39.38°	8.91°	16.9 °C	Carolea	1997–1999	3
				Picholine	1997–1998	2
**Oristano (Sardinia, OR)**	39.9°	8.62°	17 °C	Carolea	2014–2019	6
				Frantoio	2014–2020	7
**Valenzano (Apulia, BA)**	41.03°	16.85°	16.5 °C	Carolea	1997–2000	4
				Picholine	1997–2000	4
**Torre Allegra (Sicily, CT)**	37.41°	15.00°	16.6 °C	Carolea	1997 and 1999	2
				Picholine	1997 and 1999	2
**Belice Mare (Sicily, TP)**	37.60°	12.85°	18.84 °C	Carolea	1997–1998	2
				Picholine	1997–1998	2
**Rende (Calabria, CS)**	39.36°	16.23°	16.3 °C	Carolea	1999	1
				Picholine	1997 and 1999	2
**Saint Apollinare (Perugia, PG)**	43.04°	12.25°	14 °C	Carolea	1997–1999	3
				Picholine	1997–1999	3
				Frantoio	1997–1999	3
				Moraiolo	1997–1999	3
**Mirto Crosia (Cosenza, CS)**	39.72°	16.75°	18.14 °C	Carolea	2001–2003, 2015, 2018, 2019, and 2021	8
				Moraiolo	2001, 2002, 2003, 2019, and 2021	5
				Frantoio	2001, 2002, 2003, and 2019	4

**Table 3 plants-12-03181-t003:** Chill days (Cd) and anti-chill days (Ca) equations, accounting for mean (Tm), maximum (Tx), and minimum (Tn) daily temperatures and threshold temperatures (Tc).

Case	Temperature Conditions	Chill Days	Anti-Chill Days
**1**	0 ≤ Tc ≤ Tn ≤ Tx	Cd = 0	Ca = Tm − Tc
**2**	0 ≤ Tn ≤ Tc < Tx	Cd = − ((Tm − Tn) − ([Tx − Tc]^2^/2[Tx − Tn]))	Ca = ((Tx − Tc)^2^/2(Tx − Tn))
**3**	0 ≤ Tn ≤ Tx ≤ Tc	Cd = − (Tm − Tn)	Ca = 0
**4**	Tn < 0 ≤ Tx ≤ Tc	Cd = − (Tx^2^/2(Tx − Tn))	Ca = 0
**5**	Tn < 0 < Tc < Tx	Cd = − (Tx^2^/2(Tx − Tn)) − ((Tx − Tc)^2^/2(Tx − Tn))	Ca = ((Tx − Tc)^2^/2(Tx − Tn))
**6**	Tn < Tx < 0 < Tc	Cd = 0	Ca = 0

**Table 4 plants-12-03181-t004:** The root mean square error (RMSE, in days) from the calibration of the phenological models (CAC, CAC_GDD, and PhenoFlex) by phenological phase and cultivar.

Phase Olive Cultivar	Sprouting		Blooming		Pit Hardening	
	PhenoFlex	CAC	PhenoFlex	CAC_GDD	PhenoFlex	CAC_GDD
**Carolea**	22	20	14	9	40	10
**Picholine**	24	24	14	12	42	13
**Frantoio**	9.5	9	18	4	13	5
**Moraiolo**	5	6	16	4	23	6

Note: CAC, chill and anti-chill days model; CAC_GDD, chill and anti-chill days and growing degree days model.

**Table 5 plants-12-03181-t005:** Optimized parameters for the chill and anti-chill and growing degree days phenological models by phenological phase and cultivar using the genetic algorithm method.

Phenological Phase	Cultivar	CAC		CAC_GDD		
		Tc	Cr	Tb	Tx	Hr
**Sprouting**	Carolea	9.5	−115	-	-	-
	Picholine	8.2	−122	-	-	-
	Moraiolo	11.2	−133	-	-	-
	Frantoio	9.7	−137	-	-	-
**Blooming**	Carolea	-	-	5.9	31.5	437
	Picholine	-	-	7	28.3	568
	Moraiolo	-	-	4.5	29.5	370
	Frantoio	-	-	5.3	26.3	388
**Pit hardening**	Carolea	-	-	5.9	31.5	1074
	Picholine	-	-	7	28.3	1003
	Moraiolo	-	-	4.5	29.5	1315
	Frantoio	-	-	5.3	26.3	1275

Note: CAC, chill and anti-chill days model; CAC_GDD, chill and anti-chill days and growing degree days model; Tb, estimated base temperature; Tx, estimated max temperature; Hr, estimated heating requirement; Tc, estimated temperature threshold; Cr, estimated chilling requirement.

**Table 6 plants-12-03181-t006:** The results obtained from the cross-validation analysis (LOOCV) of the two phenological models for each phenological phase and olive cultivar, including the mean and standard deviation values (in days) and the number of samples/observations.

Phase Olive Cultivar		Sprouting		Blooming		Pit Hardening	
		PhenoFlex	CAC	PhenoFlex	CAC_GDD	PhenoFlex	CAC_GDD
**Carolea**	Mean	22.4	20.43	29.9	8.82	>40	9.8
	Stdv.	1.5	0.87	0.5	0.61	3	1
	Sample Number	22		21		13	
**Picholine**	Mean	26.8	24.94	36.4	12.23	>40	14
	Stdv.	1.1	1.3	4.2	0.83	1.9	0.97
	Sample Number	16		16		14	
**Frantoio**	Mean	12.48	11	31	6	>40	4.5
	Stdv.	1.4	1.5	2.2	0.4	3.35	0.5
	Sample Number	12		12		6	
**Moraiolo**	Mean	5.65	5.74	34	3.47	>40	5.6
	Stdv.	1.3	1.16	4.5	1	3.27	0.72
	Sample Number	6		6		6	

## Data Availability

https://cds.climate.copernicus.eu/cdsapp#!/search?type=dataset (accessed on 1 May 2022).

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
