# Peer review of "Modeling Phenological Phases across Olive Cultivars in the Mediterranean"

_plants, 2023, doi:10.3390/plants12183181_

Round 1

Reviewer 1 Report

The procedure that the authors used is understandable. But the conclusion is not clear.

The manuscript scientifically sounds but the experimental design is not  appropriate to test the hypothesis becouse the hypoteis is not clear.

The English languaguage needs to improve ( Line395-typography) and the punctuation mistakes should be checked  and some formatting (figure 59)

Introducction is well-organized  

The objectives of the work is not clear, and must t be clearly explaned.  Beeing the objective:“This study aims to integrate observations from several experimental sites with different environmental characteristics over the Mediterranean climate in Italy and estimate dates for three main phenological phases, including sprouting, blooming, and pit hardening of early and mid-late budbreak olive cultivars.” . –In the conclusion is not found the anwser!

 All the objectives initially  set out in the paper must be elucidated and solved in the conclusions. And this is not clear in this work.

 Which model is bether CAC_GDD or CAD_CDD?  

References must be improved, ( 13,33,35,38, 53,54,55? )

Author Response

Dear Referee,

The Journal of Plants-MDPI

 28,08, 2023

Many thanks for your time and constructive comments.

Kindly find our responses below in italic corresponding to each point.

By the way, please note that all the modifications in the manuscript`s text are highlighted in yellow background.  

#Reviewer 1:

1)The procedure that the authors used is understandable. But the conclusion is not clear.

We want to thank the reviewer for positive and useful feedback. We have modified the conclusion to

make it as clear as possible. To this end, and following other reviewer’s comments we also added two new

paragraphs in the conclusions (Lines 612-621 and 644-649). 

2)The manuscript scientifically sounds but the experimental design is not appropriate to test the hypothesis because the hypothesis is not clear.

In order to make the hypothesis clearer, we added the following paragraph at the end of our introduction (before the last paragraph of the introduction):

“Modelling phenological stages of olive trees in regional or global scales (e.g., Mediterranean basin) is highly challengeable due to limited and scattered observations in heterogenous environments. Some complex phenology models, qualified for a single phenological stage, e.g. Phenoflex that was built upon many parameters, dependent on long-term phenological observations and hourly temperature time series, would miss to project comprehensively reliable phenological dates in our above-mentioned experimental conditions. So, a general approach with less parameter demanding structure and flexible projection capability would perform better modelling successive phenological phases of olive crop over a Mediterranean environment, and assist producers, particularly in a changing climate context, with crop cultivation management, planning crop practices in orchards, and mitigating climate-induced risks.”

3)The English language needs to improve ( Line395-typography) and the punctuation mistakes should be checked  and some formatting (figure 59).

We checked and solved some punctuation mistakes.

4)Introduction is well-organized 

Thanks again for the nice and encouraging feedback

5)The objectives of the work is not clear, and must be clearly explained.  Being the objective: “This study aims to integrate observations from several experimental sites with different environmental characteristics over the Mediterranean climate in Italy and estimate dates for three main phenological phases, including sprouting, blooming, and pit hardening of early and mid-late budbreak olive cultivars.” . –In the conclusion is not found the answer! All the objectives initially set out in the paper must be elucidated and solved in the conclusions. And this is not clear in this work.

Yes, our main research objective was not very clear to the point and straightforward. We have re-elaborated and simplified it to make it more clear in the last paragraph of the introduction as follows “ This study aims to model and validate simulated dates for three main phenological phases, including sprouting, blooming, and pit hardening of early and mid-late budbreak olive cultivars, over the Mediterranean climate. To this end, we integrated observations from several experimental sites with different environmental characteristics in Italy, and framed an innovative method to combine chilling and sequentially forcing per phenological phases.”

Furthermore, we elaborated and included an additional paragraph in conclusion as answer to the objectives stated in the introduction, as suggested by the reviewer:

“Two approaches applying chilling and forcing-based models were compared to determine the performance for predicting olive phenology. This investigation presented a combined method (CAC_GDD) to estimate olive phenological phases and compared results using a more complex and data demanding model (PhenoFlex). Under our experimental conditions, over a large Mediterranean environment with scarce and scattered observations, the CAC_GDD model, with lower parameter demand and simple approach demonstrated more reliable performance than the PhenoFlex model to generalize projections at regional scale in at least two phenological phases, i.e., the blooming and pit hardening stages. However, in terms of cultivar type both models performed relatively better for the mid-late than the early budbreak cultivar.”

6)Which model is bether CAC_GDD or CAD_CDD?

We found that the CAC_GDD model/our approach can simulate the phenological stages of olive more easily and effectively, comparing to Phenoflex model, under the constraints present for our available data/simulations. The detailed advantages from CAC_GDD modelling over PhenoFlex are mentioned in the second paragraph of the conclusion.  

7)References must be improved, ( 13,33,35,38, 53,54,55? ).

We tried to review the formatting of the references.  Indeed, the mentioned references are quite relevant. By the way, we cited only 52 references in the manuscript, indeed a few of the mentioned references (53-55) were just wrongly numbered, please ignore them.  

Thank you again for reviewing our manuscript, and we hope our responses could address your concerns.

Sincerely,

Ali Didevarasl

PhD researcher at the Department of Agricultural Sciences,

The University of Sassari,

Sassari, Italy

Reviewer 2 Report

In this manuscript, the authors tested the performances of two phenological models using the phenological observation data of four olive varieties in nine locations in Italy, and the results show that the CAC_GDD model can simulate the phenological period of olives more easily and effectively. The organization of the manuscript is complete, and the method applied is suitable, and the results are clear. I have a major concern and some minor points need to be addressed by the authors for future revision:

My major revision is as the following:

As you evaluated the performances of different models for different olive cultivars, could you please tell us to what extent do the environmental requirements differ between cultivars for different phenophases? Is the difference mainly caused by different biological traits or caused by environmental factors? If the problem is difficult, please conduct some discussions at this.

I have also some minor points.

(1) Line 41-43, this viewpoint sounds a little bit arbitrary. If there are any other influencing factors, such as soil characteristics, for plants growing processes?

(2) According to Table 1 and Table 6, compared with the number of parameters of the model (especially the Phenoflex model), the data sample size used in the study was not so large. So, I am wondering if the sample size in this study was large enough to support the calculation of parameters and the reliability of the results?

(3) In line 434, you mentioned "Both models performed better for the mid-late budbreak cultivars than the early ones, regardless of the phenological stages." Could you give a detailed and reasonable explanation at this?

(4) In line 555, while considering that a warmer environment can accelerate the accumulation of forcing units, the negative effects of warmer environments on chilling accumulation should also be considered.

(5) You said your aim was clear and your aim was “to integrate observations from several experimental sites with different environmental characteristics over the Mediterranean climate in Italy and estimate 110 dates for three main phenological phases”. However, it will be better to further tell the future application directions of your discoveries in this study.

Author Response

Dear Referee,

The Journal of Plants-MDPI

 28,08, 2023

Many thanks for your time and constructive comments.

Kindly find our responses below in italic corresponding to each point.

By the way, please note that all the modifications in the manuscript`s text are highlighted in yellow background.  

#Reviewer 2:

In this manuscript, the authors tested the performances of two phenological models using the phenological observation data of four olive varieties in nine locations in Italy, and the results show that the CAC_GDD model can simulate the phenological period of olives more easily and effectively. The organization of the manuscript is complete, and the method applied is suitable, and the results are clear. I have a major concern and some minor points need to be addressed by the authors for future revision:

Thanks for the positive feedback

My major revision is as the following:

As you evaluated the performances of different models for different olive cultivars, could you please tell us to what extent do the environmental requirements differ between cultivars for different phenophases? Is the difference mainly caused by different biological traits or caused by environmental factors? If the problem is difficult, please conduct some discussions at this. 

Indeed, different environmental requirements, in our research including chilling and forcing requirements which are dependent on temperatures would differ by cultivars and the phenological stages as you mentioned correctly and this finding is also clear in our parametrization results showing different amount of requirements for each cultivar. But as mentioned already in the text due to limited observations in each site we integrated them to run model calibration, accordingly we could not test the chilling or heating requirements of same cultivar by sites to find out whether the requirements would differ by environmental factors/climate/temperatures!

Honestly, it would hard to elaborate given the limited available data (and far from the aim of the work). Some speculations may be elaborated in the discussion like inherence to genetic factors: 1) growth related gene's expression; 2) morphological traits. For instance, early varieties may produce tissues more resistant to higher temperatures during flowering and pit hardening. As mentioned these evaluations may be however speculative, and out of the scope of this work.  

By the way, as per your suggestion we tried to add complementary explanations in the discussion, highlighted in lines 445-449 and particularly lines 468-486. Hope they will address your concern.  

I have also some minor points.

1)Line 41-43, this viewpoint sounds a little bit arbitrary. If there are any other influencing factors, such as soil characteristics, for plants growing processes?

Sure, there are many parameters influencing plants growth (e.g. soil characteristics, distance from the sea, photoperiod, latitude, topography, rainfall) they are mentioned in our text also in lines 64-66. But we added in the text where you mentioned the following sentence to get it as much complete as possible: “however, still there are other ecological parameters affecting suitability of olive growth in different environments, e.g., soil characteristics”

2) According to Table 1 and Table 6, compared with the number of parameters of the model (especially the Phenoflex model), the data sample size used in the study was not so large. So, I am wondering if the sample size in this study was large enough to support the calculation of parameters and the reliability of the results?

As you pointed out correctly the sample size/number is limited and not equal for different phenological stages as well as different olive cultivars. This is the challenge we face anyway in the Mediterranean environment when gathering phenological observations for many tree crops including olive. Indeed, our approach was built to address such challenge, by presenting a general model with less parameter/data demanding structure which even performs reasonably well using limited number of observations. Our findings showed that using a limited sample size the CAC_GDD model performs relatively better than phenoFlex model.  However, providing large sample size may result in more reliable findings, as we are planning to test this hypothesis for other crops, e.g., grape in our next project.              

3) In line 434, you mentioned "Both models performed better for the mid-late budbreak cultivars than the early ones, regardless of the phenological stages." Could you give a detailed and reasonable explanation at this?

Indeed, referring to the results/RMSE values from the calibration and validation analysis we verified that the both models performed better for the late cultivars. The details indicating the obtained RMSEs are clearly written in lines 303-310 (first paragraph of the result section) and in the tables 4 and 6.

Specifically In the discussion (lines 440-446), we have mentioned some explanations for the reasons why phenological dates of late cultivars can be better simulated by the applied models as follows: “Moreover, mid-late cultivars showed similar behavior or parameters, in threshold temperatures (Tc, Tb, and Tx) and Hr. In contrast, Carolea and Picholine, selected as early budbreak cultivars showed different discrepancies between each other. These findings highlight the importance to consider the varietal factor to perform reliable and reproducible phenological models. It is worth noting that Moraiolo and Frantoio, as mid-late cultivars, are native to the same area (Central Italy), while this is different in the case of Carolea (South Italy) and Picholine (France) [42, 43]“ .

4) In line 555, while considering that a warmer environment can accelerate the accumulation of forcing units, the negative effects of warmer environments on chilling accumulation should also be considered.

Indeed, chilling accumulation depends on the defined temperature threshold/Tc. So, if daily temperatures don`t exceed the threshold during endo-dormancy period, chill requirement would be fulfilled earlier even in warmer climates, plus faster accumulation of forcing, the phenological dates will be estimated earlier in warmer environment as mentioned in the text. To make it clearer we tried to add a complementary explanation in the text as well (lines 592-593). 

(5) You said your aim was clear and your aim was “to integrate observations from several experimental sites with different environmental characteristics over the Mediterranean climate in Italy and estimate dates for three main phenological phases”. However, it will be better to further tell the future application directions of your discoveries in this study.

Concerning the future application of our approach, we included additional paragraph in the conclusions (lines 644-649) as follows:

“Our approach will support olive producer’s response to future climate change through resilient strategic management of olive cultivation, varietal choice, cultural practices, and mitigating climate-induced risks based on the reliable projections of different phenological dates. As the future research vision, the CAC_GDD model can support scaling up spatially to large region, e.g. Euro-Mediterranean, to display the environmental differences in phenological projections under future climate change scenarios.”

Thank you again for reviewing our manuscript, and we hope our responses could address your concerns.

Sincerely,

Ali Didevarasl

PhD researcher at the Department of Agricultural Sciences,

The University of Sassari,

Sassari, Italy

Round 2

Reviewer 1 Report

Before publishing it is necessary to correct the title of the tables and figures, in all the work there is no uniformity, so the journal's instructions are not followed.

Reviewer 2 Report

All my concerns have been addressed, and I have no further problem and suggestions. I recommend accept this manuscript for publication.